# BRAINS: A Retrieval-Augmented Agent for Alzheimer's Detection and Monitoring

**Md Kishor Morol**[1]                                                                       MMOROL@CORNELL.EDU
**Md Tanzib Hosain**[2]                                                        20-42737-1@STUDENT.AIUB.EDU
**Nafiz Fahad**[3]                                                          NAFIZ.FAHAD@STUDENT.MMU.EDU.MY
**Md. Jakir Hossen**[3]                                                            JAKIR.HOSSEN@MMU.EDU.MY
**Mohammad Ali Moni**[*4]                                                                 M.MONI@UQ.EDU.AU

[1] *College of Computing and Information Science, Cornell University, New York, United States*

[2] *Department of Computer Science, American International University-Bangladesh, Dhaka, Bangladesh*

[3] *Faculty of Information Science and Technology, Multimedia University, Melaka, Malaysia*

[4] *School of Health and Rehabilitation Sciences, The University of Queensland, Queensland, Australia*

**Editors:** Accepted for publication at MIDL 2026

## Abstract

As Alzheimer's disease (AD) continues to impose a growing global burden, early and accurate detection remains essential, particularly in low-resource settings. To address this challenge, we propose **BRAINS** (*Biomedical Retrieval-Augmented Intelligence for Neurodegeneration Screening*), a retrieval-augmented framework for Alzheimer's detection and monitoring. BRAINS combines a Diagnostic Module, which applies fine-tuned LLMs to cognitive and neuroimaging data such as MMSE, CDR, and brain volume measures, with a Case Retrieval Module that retrieves similar patient profiles from a curated knowledge base. Retrieved cases are integrated through a Case Fusion Layer to improve contextual reasoning before inference. Experiments on real-world datasets show that BRAINS effectively identifies early cognitive decline and classifies disease severity, highlighting its promise as a scalable, explainable tool for early-stage Alzheimer's screening.

**Keywords:** Alzheimer's Disease, Retrieval Augmented Generation, Agents, Clinical Decision Support, Small Language Models.

## 1. Introduction

Alzheimer's disease (AD), the leading cause of dementia, is a progressive neurodegenerative disorder that impairs memory, cognition, and behaviour Imbimbo et al. (2021). It remains widely underdiagnosed, particularly in low-resource settings, while early diagnosis is hindered by costly and specialist-dependent tools such as MRI-based analysis Marcus et al. (2007) and clinical scales including MMSE and CDR Morris (1993a); Miller (2018); Petersen et al. (2010); Jack et al. (2018); Weiner et al. (2015); Folstein et al. (1975). The difficulty is further amplified by subtle brain changes and the variability of clinical indicators such as eTIV, nWBV, MMSE, and CDR, motivating intelligent multimodal diagnostic agents Reuben et al. (2021); Yang et al. (2023); Singhal et al. (2023); Luo et al. (2024); Zeng et al. (2024); Gao et al. (2023); Zhang et al. (2024).

Recent large language models (LLMs) provide a flexible framework for reasoning over structured clinical data Chen et al. (2024); Chowdhery et al. (2023); Achiam et al. (2023);

---

[*] Corresponding author

Li et al. (2023); Touvron et al. (2023), but existing agents remain limited in interpretability and case-based reasoning. To address this, we propose **BRAINS** (*Biomedical Retrieval-Augmented Intelligence for Neurodegeneration Screening*), a retrieval-augmented framework that combines LLM reasoning, case retrieval, and neurocognitive data fusion. By integrating semantically relevant historical cases through a Case Fusion Layer, BRAINS enables more accurate and interpretable Alzheimer's detection and staging from cognitive, volumetric, and demographic data.

## 2. Methodology

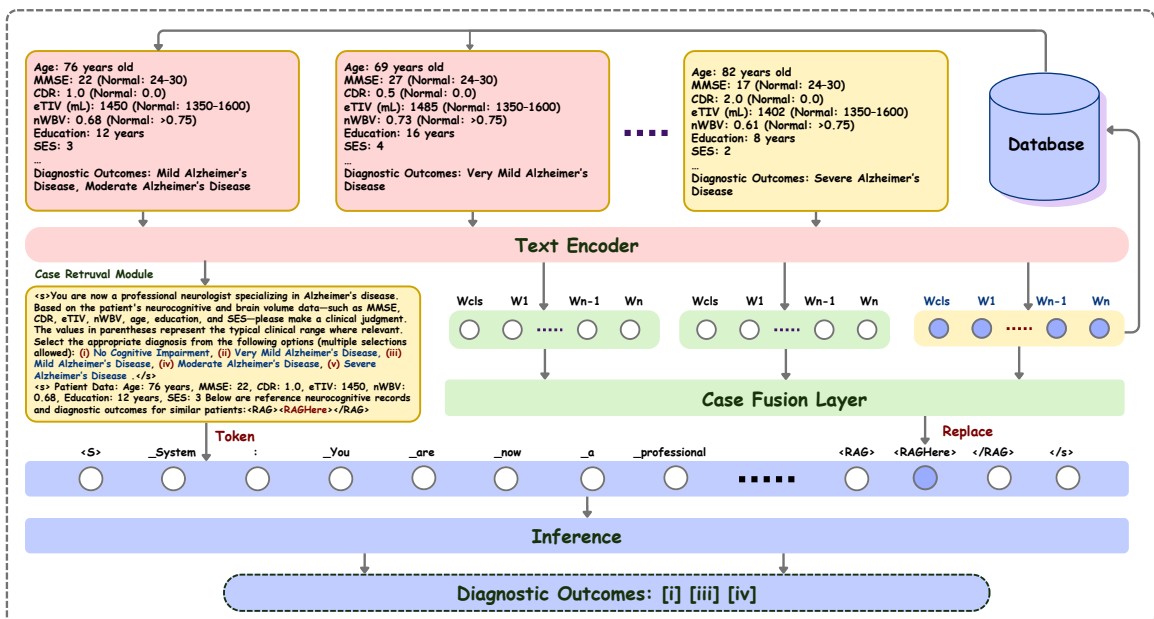

Figure 1: BRAINS architecture for Alzheimer's diagnosis. The input case is encoded and used to retrieve similar neurocognitive records. Retrieved cases are fused with the input via the Case Fusion Layer, replacing the `<RAGHere>` token in the prompt. The fused representation is then passed to the LLM for inference and explanation.

**BRAINS** is a retrieval-augmented framework for Alzheimer's disease diagnosis that combines domain-adapted language modeling with case-based neurocognitive reasoning. We pre-train the model on Alzheimer's-related reports and summaries Frisoni et al. (2010); Weiner et al. (2015); Marcus et al. (2007), neurocognitive evaluations such as MMSE and CDR Morris (1993b), and structured annotations from NACC and ADNI (NACC,A). Because the model is text-only, visually referential sentences are removed. For downstream evaluation, we use a clinical dataset of 1105 patient records containing MMSE, CDR, eTIV, nWBV, age, gender, handedness, education, and socioeconomic status, with preprocessing through normalization, encoding, and outlier removal.

BRAINS includes a *Case Retrieval Module* and a *Diagnostic Module*. Cases are encoded and stored in a FAISS vector database Douze et al. (2025); for each input, the retriever reranks similar historical cases and selects the top-$K = 5$ auxiliary profiles. The Diagnostic Module then fuses retrieved and target representations through Transformer-based cross-attention Vaswani et al. (2017), generating a retrieval-aware representation for infer-

ence. Following established neuroscience benchmark practices Guo et al. (2024), we adopt `LLaMA2-13B` as the backbone Touvron et al. (2023). Pre-training runs for 10 epochs with batch size 64, AdamW Loshchilov and Hutter (2017), learning rate $1 \times 10^{-4}$, 1,000 warm-up steps, and block size 2048. Fine-tuning uses the same encoder, `bge-reranker-large` for reranking Xiao et al. (2023), and LoRA with $\alpha = 32$ and $r = 8$ Hu et al. (2022), trained for 15 epochs with batch size 4, AdamW Loshchilov and Hutter (2017), learning rate $1 \times 10^{-5}$, and dynamic masking over $m \in [0, 4]$ retrieved cases. Pre-training uses next-token prediction, whereas fine-tuning optimizes supervised loss only on the assistant response.

## 3. Results

Table 1: Performance comparison of `LLaMA2-13B`, RAG variants, and the proposed BRAINS agent across all, single, double, and triple case types.

| Model | All | | Single | | | Double | | | Triple | | |
|---|---|---|---|---|---|---|---|---|---|---|---|
| | Correct | F1 | Prec. | Recall | F1 | Prec. | Recall | F1 | Prec. | Recall | F1 |
| **LLaMA2-13B** | | | | | | | | | | | |
| Five-shot | 0.335 | 0.339 | 0.000 | 0.000 | 0.000 | 0.299 | 0.719 | 0.423 | 0.421 | 0.980 | 0.591 |
| Fine-tuning | 0.600 | 0.538 | 0.657 | 0.728 | 0.692 | 0.468 | 0.474 | 0.471 | 0.643 | 0.281 | 0.391 |
| w/o standard | 0.454 | 0.376 | 0.645 | 0.513 | 0.571 | 0.290 | 0.500 | 0.361 | 0.250 | 0.063 | 0.100 |
| **RAG** | | | | | | | | | | | |
| RAG-1 | 0.712 | 0.731 | 0.766 | 0.540 | 0.619 | 0.703 | 0.824 | 0.802 | 0.774 | 0.981 | 0.863 |
| RAG-2 | 0.727 | 0.755 | 0.790 | 0.572 | 0.664 | 0.660 | 0.921 | 0.769 | 0.727 | 0.975 | 0.842 |
| **BRAINS** | **0.773** | **0.819** | **0.784** | **0.731** | **0.740** | **0.711** | **0.875** | **0.810** | **0.931** | **0.911** | **0.929** |

Table 1 confirms the effectiveness of BRAINS for neurocognitive disorder inference. Although `LLaMA2-13B` with Five-shot prompting generates plausible outputs, it performs unreliably on complex multi-label cases. Fine-tuning on structured clinical text with MMSE, CDR, and MRI-derived volumetric features improves accuracy by **26.50%**, while removing these biomarkers causes clear performance degradation. Retrieval augmentation further increases accuracy from **60.00%** to **71.20%** with one retrieved case, but adding more than two cases is limited by context length. By using case fusion, BRAINS overcomes this issue, integrates up to five auxiliary cases, and achieves **77.30%** accuracy.

With Five-shot prompting, the model attains high recall (**98.00%**) but low precision, yielding an F1 score of only **59.10%** in multi-pathology settings; for single-label prediction, performance collapses (**F1 = 0.00%**). Fine-tuning reduces this imbalance, while BRAINS provides the most robust, interpretable, and accurate predictions across varying diagnostic complexity.

## 4. Conclusion

This study introduces **BRAINS**, a foundation model for early-stage Alzheimer's screening. Designed to analyse neurological report data—including MMSE, CDR, speech and behaviour logs, and structural brain imaging summaries—BRAINS supports clinical reasoning, particularly for less experienced practitioners. By integrating retrieval-augmented generation (RAG), it improves diagnostic precision in multi-morbidity inference tasks. In benchmark evaluations for mild cognitive impairment and Alzheimer-type dementia classification, BRAINS achieves **77.30%** accuracy, substantially outperforming the baseline large language model at **45.40%**. These results highlight BRAINS as a scalable, interpretable, and data-efficient framework with potential for broader neurological diagnostic applications.

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
