# OpenReview forum: "BRAINS: A Retrieval-Augmented Agent for Alzheimer's Detection and Monitoring"
_MIDL.io/2026/Short_Papers — MIDL 2026 - Short Papers Poster_

### Official Review · Reviewer_34uq · 2026-04-24
**Interesting method with open questions.**

**Rating:** 4
**Confidence:** 3

**Review:**

see strengths and weaknesses.

**Summary:**

The paper presents an agentic framework that integrates clinical data with historical case information for Alzheimer’s disease diagnosis. The method constructs a database of encoded prior cases, then encodes an input case to retrieve the most similar instances. Information from these retrieved cases is fused with the input to form a representation, which is subsequently used to generate the final diagnosis.

**Strengths:**

The overall approach appears technically reasonable. The authors demonstrate awareness of data preprocessing considerations—for example, removing visually referential sentences to accommodate a text-only model. The paper also provides initial experimental results, which appear promising.

**Weaknesses:**

The paper does not address how the approach could be extended beyond text-based information, which may limit its applicability in multimodal clinical settings.
The data flow illustrated in Figure 1 lacks clarity; for instance, components such as the “input case” mentioned in the caption are not clearly depicted in the figure itself.
The method mentioned "explanation" but did not provide further information.
Comparing with baselines that are not based on LLM is encouraged.

**Justification Of Rating:**

The paper proposes a reasonable approach by leveraging a retrieval-augmented LLM framework to incorporate historical clinical cases into Alzheimer’s disease diagnosis. Such agentic and retrieval-based strategies are of growing interest to the community, particularly for applications that require grounding model outputs in prior knowledge or evidence.
However, the paper does not fully convince the reader of the practical benefit of this relatively complex framework in the specific context of Alzheimer’s diagnosis.

---

### Decision · Program_Chairs · 2026-05-08

Accept (Poster)